# Epithelial Ovarian Cancer—Varied Treatment Results

**DOI:** 10.3390/healthcare11142043

**Published:** 2023-07-17

**Authors:** Sonja Millert-Kalińska, Marcin Przybylski, Dominik Pruski, Małgorzata Stawicka-Niełacna, Radosław Mądry

**Affiliations:** 1Doctoral School, Poznan University of Medical Sciences, 61-701 Poznań, Poland; 2Department of Obstetrics and Gynecology, District Public Hospital in Poznan, 60-479 Poznań, Polanddominik.pruski@icloud.com (D.P.); 3Department of Clinical Genetics and Pathology, University of Zielona Góra, 65-046 Zielona Góra, Poland; 4Department of Oncology, Poznan University of Medical Sciences, 60-569 Poznań, Poland

**Keywords:** epithelial ovarian cancer, EOC, disparities in treatment, laparoscopy in oncology, cytoreduction

## Abstract

Ovarian cancer (OC) is the eighth most common cancer worldwide and is usually diagnosed in advanced stages. Despite many available data, no treatment results have been reviewed in Poland. This study enrolled 289 first-time patients treated between 2018 and 2021 by the Department of Oncology of the Poznań University of Medical Sciences (SKPP). The relationships among starting treatment in our centre, the type of first intervention, and the final decision were significant (*p* < 0.001). Patients in the SKPP group were more likely to primarily have a laparoscopy and less likely to have an exploratory laparotomy. Neoadjuvant chemotherapy (NACT) after a laparotomy was less often a final decision among SKPP patients (9% vs. 22%), in contrary to NACT after a laparoscopy (23% vs. 4%). Factors affecting the shortening of progression-free survival (PFS) were an advanced stage of the disease, a histopathological diagnosis, the type of cytoreduction, and the final decision. Significance according to the final decision was revealed for PDS vs. NACT after a laparotomy (*p* < 0.001) and for PDS vs. NACT after a laparoscopy (*p* = 0.011). Our study supports the benefits of treating ovarian cancer in an oncology centre with a high patient throughput. Further observations might also answer the question about overall survival (OS).

## 1. Introduction

Epithelial ovarian cancer (EOC) most often manifests itself in an advanced stage and is the second most common cause of death in women from gynaecological malignancies after cervical cancer. According to GLOBOCAN, estimates suggest that in 2020, over 300,000 new cases of ovarian cancer were documented and over 200,000 deaths were confirmed [1,2].

As reported by the National Cancer Registry, which is published annually in Poland, in 2020, OC was the second most common gynaecological cancer (4.1%) after endometrial cancer. In contrast, it was the most common cause of death (6.0%) [3]. In line with the literature, about 70% of ovarian cancer cases are diagnosed in advanced stages, which are associated with a low 5-year survival rate. According to the International Federation of Gynaecology and Obstetrics (FIGO), the 5-year survival rate in patients with OC diagnosed at stages III or IV is 30–50%, whereas for those diagnosed at stages I and II, it is 80–90% [4].

Clinicians might manage therapy by considering the time to relapse and the toxicity profile associated with the side effects of previous lines of treatment. The proposed regimens require attending physicians to be prudent based on their experience. Therefore, the long-term monitoring of patients continuing treatment in one centre is essential. Multiple population-based studies have documented the relationship between treatments in high-volume hospitals and the survival rate of patients with ovarian cancer, showing that superior treatment and survival outcomes are associated with multidisciplinary resources offered by physicians and hospitals.

The available databases are up-to-date and present the epidemiological situation of patients with ovarian cancer in Poland. However, no publications have examined the treatment results for this cancer in Poland. Our paper discusses the treatment results of patients with epithelial ovarian cancer in one centre with those treated in different hospitals.

## 2. Materials and Methods

### 2.1. Study Design

The study received approval from the Poznań University of Medical Sciences’ bioethics committee regarding the lack of a medical experiment and its retrospective nature. We present a retrospective observational study for the clinical hospital of the University of Medical Sciences in Poznań, Poland (abbreviated to SKPP). The material consisted of patients treated in the Department of Gynaecological Oncology due to ovarian cancer. We conducted a detailed, multifaceted analysis of patients admitted to the ward for the first time in 2018–2021. We aimed to determine the FIGO profile of patients, the scope of the first intervention, and the final decision. Finally, we analysed the progression-free survival (PFS).

Our study initially analysed patients referred to the Department of Gynaecological Oncology ward in 2018–2021 with suspected ovarian cancer. The inclusion criteria were as follows: (i) patients with a histopathological confirmation of ovarian cancer based on surgery with a laparotomy, laparoscopy, or biopsy; (ii) patients over 18 years of age; (iii) pregnancy; and (iv) patients that provided written consent for the proposed treatment, including both the surgical treatment and the subsequent chemotherapy. The exclusion criteria were: (i) patients with another confirmed histopathological diagnosis, including intestinal tumours, tumours of the genital organs not derived from the ovary, and benign lesions, and (ii) patients transferred from other centres to obtain subsequent lines of chemotherapy treatment. Patients who did not qualify for a specific type of chemotherapy due to their poor general condition or who did not accept the proposed treatment were rejected. Additionally, patients who discontinued a course of chemotherapy were excluded from the study. The final exclusion criterion was patients undergoing fertility-conserving treatment (FCT).

### 2.2. Statistical Analysis

All analyses were conducted using R statistical software, version 4.2.1. Nominal variables are presented in the tables as the number and percentage of observations. Quantitative variables were described with a median and range of scores (normality of distribution was checked with a Shapiro–Wilk test; all parameters were proven to have a non-normal distribution, with *p* < 0.05). Dependencies between two variables were analysed using a chi-squared test (with Yate’s correction for continuity for 2 × 2 tables) or Fisher’s exact test, depending on whether the assumption of expected counts of not less than five in at least 80% of the cells in the respective contingency table was satisfied. The strength of significant relationships between two categorical variables was measured using Cramer’s V coefficient, with a 95% confidence interval. A series of univariate Cox regressions were conducted; hazard ratios with 95% confidence intervals and *p*-values are presented in the tables. Quantitative variables were compared between groups with Wilcoxon’s test for independent groups, due to the non-parametric nature of the analysed numeric variables. A multivariate Cox regression model was built using a step method—variables with a *p*-value less than 0.250 in the univariate models were used as predictors in the initial multivariate model [5]. One variable was discarded, as the number of observations for this variable was much lower than the number of observations for the other variables in the model. The VIF value was checked while building the final multivariate model. Relapse-free survival curves are shown for all patients and were broken down into smaller groups. Log-rank tests were conducted to determine the differences in the relapse rates between groups.

## 3. Results

In 2018–2021, 289 subjects reported to the Department of Gynaecological Oncology and fulfilled all inclusion criteria. At the time of diagnosis, 224 patients (76.9%) were admitted to the hospital with an advanced stage of the disease. For stage III of the disease, 43% of patients were diagnosed, whereas 27% had stage IV of the disease. Two-thirds of the study group started their therapeutic path in our centre, whereas 99 patients were transferred from other hospitals.

Considering the type of the first intervention performed on patients, we made the following division: (1) a PDS, or primary debulking surgery; (2) a laparoscopy; (3) an exploratory laparotomy, which resulted in a biopsy; and (4) no surgical intervention.

Patients from the last group underwent further treatment based on the results of imaging tests. Regarding the first interventions in our centre, seventy patients underwent a primary laparoscopy. Women who underwent a PDS constituted 43.2% of the cohort; a biopsy was performed in less than one-fifth of the patients, and in seventeen patients, no intervention was undertaken. When it came to patients starting treatment in other hospitals, half of them had a PDS before being transferred to the SKPP. Only 7% of women underwent a primary laparoscopy; more than 40% underwent an exploratory laparotomy. Only five patients were referred to SKPP without any primary surgical interventions.

Then, all patients were divided into groups depending on the final treatment decision as follows: (1) a PDS—primary debulking surgery—performed immediately after a laparotomy and in patients who received an exploratory laparoscopy; (2) NACT after a laparotomy, where the neoadjuvant treatment was followed by a biopsy or an adnexectomy in the laparotomy; (3) NACT after a laparoscopy, where the neoadjuvant treatment took place when the laparoscopic evaluation did not allow for complete cytoreductive surgery; (4) chemotherapy, considered in those patients who did not qualify for surgery; and (5) NACT without primary surgery, the patients of which received the neoadjuvant treatment based on imaging tests without a primary intervention.

Nearly 74% of women had histopathologically diagnosed HGSC, whereas other subtypes, such as LGSC, clear-cell carcinoma, endometrioid cancer, etc., were observed significantly less frequently. In the group of patients, we assessed the achieved cytoreduction. We considered 237 out of 291 women. Firstly, not all patients had a surgical intervention during treatment. Secondly, we did not obtain all the operating protocols for the women that were operated on in other hospitals. The stage of the obtained cytoreduction was most often assessed as total (48.5%), then suboptimal (18.6%), and, least often, optimal (32.9%). The baseline characteristics of the groups are presented in Table 1.

Among SKPP patients, there was a lower proportion of subjects in the non-staging FIGO category than among transferred patients (5% vs. 13%; *p* = 0.037). The relationships among the starting treatment in SKPP, the type of the first intervention, and the final decision were significant (*p* < 0.001 for both analyses). In the SKPP group of patients, there was a more considerable proportion of subjects that had a laparoscopy than among transferred ones (37% vs. 7%) and a smaller proportion of patients that had an exploratory laparotomy ending in a biopsy (12% vs. 40%). NACT after a laparotomy was less often a final decision among SKPP patients than among transferred patients (9% vs. 22%), and NACT after a laparoscopy was more common among SKPP patients (23% vs. 4%). The dependencies mentioned above are presented in Table 2.

Among the 2018 group, there was a smaller proportion of patients that had a PDS and a greater proportion of patients that had a laparoscopy than in all other groups (28% vs. 46% in 2019, 55% in 2020, and 47% in 2021 for PDS–laparotomy; 43% vs. 17% in 2019, 24% in 2020, and 25% in 2021 for laparoscopy; *p* = 0.003). A PDS after a laparoscopy or laparotomy as the final decision was less common in the 2019 group than in the other groups (44% vs. 64% in 2018, 58% in 2020, and 67% in 2021). NACT after a laparotomy was more common as the final decision among the 2019 group than among the other groups (29% vs. 8% in 2018, 13% in 2020, and 6% in 2021; *p* < 0.001). No significant dependency was observed between the analysed groups and the FIGO staging system, the place of the first intervention, or the stage of cytoreduction, as presented in Table 3.

The result of the operation was not related to the stage of the disease according to the FIGO classification (when analysing only stages III and IV) or to the place of the first intervention—*p* = 0.533 for FIGO and *p* = 0.060 for the place of the first intervention, as shown in Table 4.

Among subjects with an R1 or R2 stage of cytoreduction, there was a more significant proportion of patients from the group classified as FIGO stage III, stage IV, or no staging than patients from the group classified as FIGO stages I or II (98% vs. 2% for R1 and 99% vs. 1% for R2, respectively; *p* < 0.001). The operation result was unrelated to the place of the first intervention—*p* = 0.060, as shown in Table 4.

According to the National Cancer Institute, relapse is the return of a disease or the signs and symptoms of a disease after a period of improvement, and progression-free survival (PFS) is the length of time during and after the treatment of a disease that a patient lives with the disease, but does not experience a worsening of the disease. The time from the second to the third relapse was significantly longer for patients with stage IV of the disease according to the FIGO classification (Me = 5.00 months) than for patients with stage III (Me = 3.00 months)—*p* = 0.009. No other differences between the analysed groups were observed, as shown in Table 5.

The values are shown as the median with a range of scores. The comparisons were computed using Wilcoxon’s test for independent groups.

Almost all univariate Cox regression models for the occurrence of a relapse were significant. Patients with the disease in stage III, stage IV, or no staging had an 8.49 times higher risk of a relapse compared to patients with the disease in stages I or II (95% CI for HR = 3.12–23.10; *p* < 0.001). Serous high-grade histopathology results decreased the risk of relapse by 74% (95% CI for HR = 0.14–0.51; *p* < 0.001). Patients with a laparoscopy or an exploratory laparotomy had a 2.28- and 1.92-times higher risk of relapse, respectively, than those who underwent a primary debulking surgery in a laparotomy (95% CI for HR = 1.38-3.59 and *p* = 0.001 for laparoscopy; 1.14–3.24 and *p* = 0.015 for exploratory laparotomy). Subjects whose final decision was NACT after a laparotomy or a laparoscopy had a 2.97- and 2.52-times higher risk of relapse, respectively, than subjects who underwent a PDS as the final decision (95% CI for HR = 1.78–4.52 and *p* < 0.001 for NACT after laparotomy; 1.53–4.14 and *p* < 0.001 for NACT after a laparoscopy). The result of the operation was the last significant predictor for the occurrence of a relapse—stage R1 compared to stage R0 increased the risk of a relapse by 4.02 times (95% CI for HR = 2.23–7.25; *p* < 0.001) and stage R2 compared to stage R0 increased that risk by 3.55 times (95% CI for HR = 2.08–6.05; *p* < 0.001), as presented in Table 6.

The multivariate Cox regression model was built using a step method. All variables from the significant univariate models were put into the initial multivariate model. The variables that were left in the final multivariate model were: the degree of advancement of the disease (FIGO; *p* = 0.071), the histopathology (*p* = 0.079), the final decision (PDS vs. NACT after laparotomy, *p* < 0.001; PDS vs. NACT after laparoscopy, *p* = 0.011), and the stage of the cytoreduction (R1 vs. R0, *p* = 0.005; R2 vs. R0, *p* = 0.020). The FIGO stage and the histopathology variables were not significant in the multivariate model. Subjects whose final decision was NACT after a laparotomy or laparoscopy had a 2.81- and 1.99-times higher risk of relapse, respectively, than subjects who underwent a PDS as the final decision (95% CI for HR = 1.58–4.98 and *p* < 0.001 for NACT after a laparotomy; 1.17–3.39 and *p* = 0.011 for NACT after a laparoscopy). Stage R1 of cytoreduction compared to stage R0 increased the risk of relapse by 2.43 times (95% CI for HR = 1.30–4.56; *p* = 0.005), and stage R2 compared to stage R0 increased that risk by 1.96 times (95% CI for HR = 1.11–3.46; *p* = 0.020), as shown in Table 7.

The progression-free survival rates are shown in Figure 1, Figure 2, Figure 3, Figure 4, Figure 5, Figure 6, Figure 7 and Figure 8 (for all patients and broken down into smaller groups). The last patient relapsed after 46 months. The cumulative relapse-free survival rate at the end of the follow-up period was 56% for all patients. The relapse rate was not significantly different between the SKPP and transferred patients—the cumulative proportion of patients who did not relapse at the end of the follow-up was 52% for SKPP patients and 64% for other patients, as shown in Figure 2. Patients with a FIGO score of I or II had a lower relapse rate than patients with a score of III or IV, and patients with a FIGO score of II also had a higher relapse rate than patients with a FIGO score of I (*p* < 0.001). The cumulative proportion of patients who did not relapse at the end of the follow-up was 98% for a FIGO score of I, 57% for a FIGO score of II, 41% for a FIGO score of III, and 50% for a FIGO score of IV, as presented in Figure 3. The cumulative proportion of patients who did not relapse at the end of the follow-up was 46% for patients with serous HG and 83% for other patients. The relapse rate significantly differed between groups (*p* < 0.001), as presented in Figure 4. Patients from the R1 group had the highest relapse rate, followed by patients from the R2 group, and patients from the R0 group had the lowest relapse rate (*p* < 0.001). The cumulative proportion of patients who did not relapse during the follow-up was 71% for the R0 group, 33% for the R1 group, and 39% for the R2 group, as presented in Figure 5. Patients with positive and negative genetic test results did not differ significantly in their relapse rate (*p* = 0.080). The cumulative proportion of patients who did not relapse at the end of the follow-up was 42% for those with negative results and 16% for those with positive results, as shown in Figure 6. The relapse rate was higher for patients from the laparoscopy and biopsy–laparotomy groups than those from the PDS group (*p* = 0.006). The cumulative proportion of patients who did not relapse at the end of the follow-up period was 67% for the PDS group, 35% for the laparoscopy group, 56% for the biopsy–laparotomy group, and 62% for patients with no operation, as shown in Figure 7. Patients from the chemotherapy and NACT-without-primary-surgery groups had the lowest relapse rate, followed by patients from the PDS group. Patients from the NACT-after-a-laparoscopy and laparotomy groups had a higher relapse rate (*p* < 0.001). The cumulative proportion of patients who did not relapse during the follow-up was 60% for the PDS group, 40% for the NACT-after-a-laparotomy group, 43% for the NACT-after-a-laparoscopy group, and 64% for the chemotherapy and NACT-without-primary-surgery groups, as presented in Figure 8.

## 4. Discussion

Our study presents the results of epithelial ovarian cancer treatment in patients referred to the Clinical Hospital of the University of Medical Sciences in Poznań in 2018–2021. The preliminary analyses concerned comparisons between the therapy results of patients treated from the beginning at the SKPP and those of patients referred initially from other medical centres. The observations of our study group are consistent with the epidemiology of ovarian cancer. Prognostic factors are features correlating primarily with survival. In OC, these features include the FIGO staging, the grading, the histological type of cancer, the clinical status of the patient at the time of the diagnosis, and the extent of primary cytoreduction.

Determining the clinical stage of FIGO enables the correct and most advantageous qualification for appropriate therapy [6]. Grading is an important supplement to a histopathological diagnosis, and additional immunohistochemical and molecular analyses allow different types of ovarian cancer to be distinguished. The cancer biology in older patients contributes to a worse prognosis, less invasive surgical treatments, and less toxic chemotherapy [7,8].

The data showed significant differences in the first step of managing a patient upon admission when a diagnosis of ovarian cancer was suspected. There was a significantly higher percentage of laparoscopic procedures in the group of patients starting treatment at the SKPP (37% vs. 7%). In contrast, fewer patients underwent an exploratory laparotomy for a biopsy and histopathological confirmation (2% vs. 40%). These data are also associated with another dependence—a higher percentage of patients undergoing NACT had a laparotomy rather than a laparoscopy. A possible explanation for this relationship is the availability of highly qualified specialists experienced in endoscopic treatment, a broader spectrum of diagnostic possibilities, and fewer emergency admissions.

The subsequent analyses focused on the final treatment results of the entire study group and observations of the length of progression-free time, and attempted to answer questions about the factors influencing the extension of these periods. Interestingly, the time to first progression in FIGO stage III and IV patients was the same at seven months. However, the analysis indicated that the time to progression was significantly longer in more advanced patients (five months in FIGO stage IV patients vs. three months in FIGO stage III patients). Further follow-up studies should be performed due to the study group’s short period of total observation.

In addition, the study confirmed commonly known facts. The stage of the disease, histopathologically confirmed HGSC, and the degree of cytoreduction obtained during the first operation were crucial regarding the impact on the recurrence time. The result regarding cytoreduction may be surprising—patients with an R1 surgery had a 4-fold higher risk of recurrence than patients with complete cytoreduction (R0). In contrast, patients with radical R2 surgery had 3.5 times more frequent recurrences.

The results are partially consistent with du Bois’s claims, who conducted an extensive analysis that included 3126 patients. A multivariate analysis showed improved progression-free times and overall survival rates for a group with complete resection (R0) compared to groups with an R1 or R2 resection (*p* < 0.0001). The impact of debulking in the R1 resection group showed a smaller prognostic impact than in the R2 resection group.

As for the primary cytoreductive operation, its importance is already undeniable. In women with R1 residual disease followed by standard chemotherapy, the mean time to recurrence was 22 months, and the overall survival was 52 months. In comparison, women with residual disease greater than 1 cm had a median time to recurrence of 14 months and an overall survival of 26 months [9,10,11,12]. In addition, studies conducted by Wimberger, Bristow, Du Bois, Chi, and others showed a direct effect of maximal cytoreduction without leaving residual disease (R0 or R1) on the PFS and OS [13,14,15,16,17]. In advanced ovarian cancer, primary surgery is the most important, and surgery should be preceded by a thorough analysis, including imaging and biochemical diagnostics. Winter et al. analysed the results of the GOG studies #111, #132, #152, and #162, in which patients were operated on suboptimally (disease remnants > 1 cm), and their survival was statistically significantly worse than in those women where the surgery was maximally cytoreductive. Du Bois achieved similar results in the three prospective studies AGO-OVAR #3, #5, and #7 [15,16,17,18].

According to the literature, the percentage of complete cytoreductive procedures achieved in the leading gynaecological oncology centres reaches 60%, and the percentage of optimal cytoreductive practices is 80%.

One can risk the statement that a further increase in the percentage of patients operated on effectively and efficiently may not be possible. This is related to the predictive factors described earlier. Appropriate patient qualification is necessary to avoid burdensome, unnecessary, and incomplete surgical procedures. The assessment of resectability by a laparoscopy, which has been developing in recent years, certainly raises hope.

However, what needs to be mentioned, and what sometimes seems to be the most challenging decision for the operator, is the definition of non-operational cases. The above conclusion may also be a generalization of our results. Namely, the progression-free time is similar in the PDS, CTH, and NACT groups without a subsequent surgery. It is longer than in the groups where surgery was performed after NACT.

In 2010, the results of a prospective randomized study were published comparing primary cytoreduction with interval cytoreduction in patients with stage III or stage IV of the disease. Interestingly, the results obtained in both groups were the same. The benefit of using IDS was the achievement of a higher percentage of patients operated on completely. However, a postponed surgery did not improve the outcome. Another study, CHORUS, also confirmed these results. Deferred surgical procedures should not be prioritized when planning treatment, but should only be considered during the so-called minimal therapeutic gain resulting from a radical procedure [16,19,20,21,22,23].

The results mentioned above confirm that the quality of a cytoreductive operation translates into the prognosis and is the only surgeon-dependent prognostic factor. This implies that only experienced operators should perform cytoreductive procedures. These results should be considered when creating ovarian cancer units, similarly to breast cancer units.

In conclusion, the importance of the good organization, speed, and efficiency of actions for suspected or confirmed ovarian cancer should be emphasized. Ovarian tumours are pretty common in gynaecological practice, and each patient with a suspected tumour should be referred to oncology centres for further diagnostics as soon as possible. Currently, the widespread lack of continuity of diagnostics and the treatment or the neglect of specific procedures deprives patients of the chance for optimal therapy, e.g., qualifying for a treatment program with bevacizumab. It is essential to precisely describe the scope of the procedures or the results of imaging tests to determine whether a patient meets the conditions of drug programs, which are very specific and strict. This requires the excellent awareness, knowledge, and cooperation of physicians in various specialities at every stage of treatment.

The results, the severity of the disease, the patient’s general condition, a delayed start to treatment, the FIGO stage, and limited access to some operating equipment and drug programs were all affected by the COVID-19 pandemic. Over the next decade, the negative direct and indirect impacts will undoubtedly affect morbidity and mortality rates worldwide. Many hospitals received patients with advanced cancer and respiratory system complications, which precluded endoscopic techniques, as in patients after thoracic surgery.

### 4.1. Strengths of the Study

The available databases are up-to-date and present the epidemiological situation of patients with ovarian cancer in Poland. However, only a few publications have examined the treatment results for this cancer in Poland. Our paper discusses the treatment results of patients with epithelial ovarian cancer in a homogenous one-centre population.

### 4.2. Limitations of the Study

The accuracy and completeness of the data in a retrospective analysis depend on the quality and availability of the existing records or databases. In some cases, the data were incomplete or lacked important variables. This limited the ability to assess certain factors or confounding variables that may have been relevant to the analysis. The reliability of the results was also limited by the short period of observation. This implies that the results cannot be used to assess the overall survival rates or 5-year survival rates. However, it will be possible to draw more conclusions due to constant follow-up and patient access.

## 5. Conclusions

Our study supports the benefits of treating ovarian cancer in an oncology centre with a high patient throughput. It supports a higher percentage of laparoscopies, fewer exploratory laparotomies, and longer PFS times in patients treated using the current medical knowledge. During a PDS, maximum cytoreduction with the expected resectability at R0 is the most critical factor affecting the PFS. Further observations will also answer questions about the OS. Future challenges include universal adherence to the same rules and the correct quality of preoperative diagnostics and real medical consultations. In addition, the importance of the quality of surgical treatment in ovarian cancer and the accuracy and quality of histopathological examinations should be emphasized.

## Figures and Tables

**Figure 1 healthcare-11-02043-f001:**
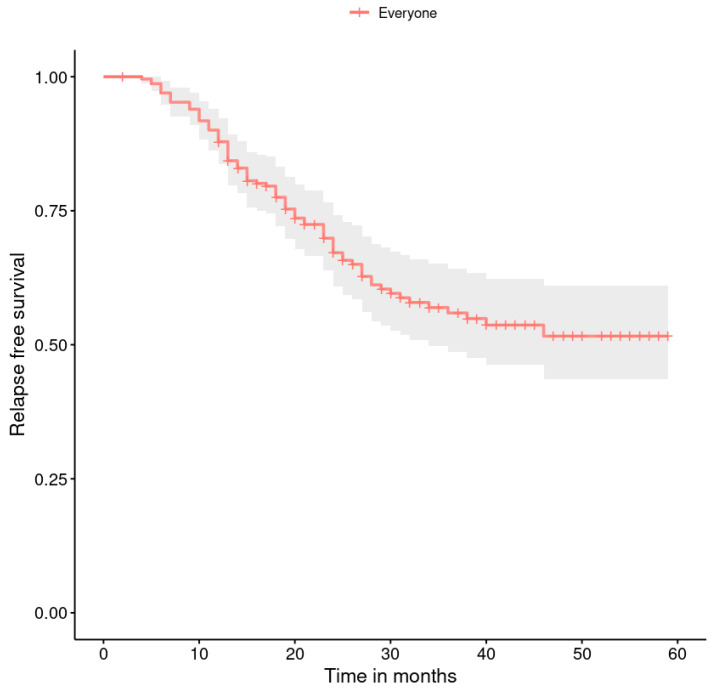
Relapse-free survival curve with 95% CI (darkened area) for all patients.

**Figure 2 healthcare-11-02043-f002:**
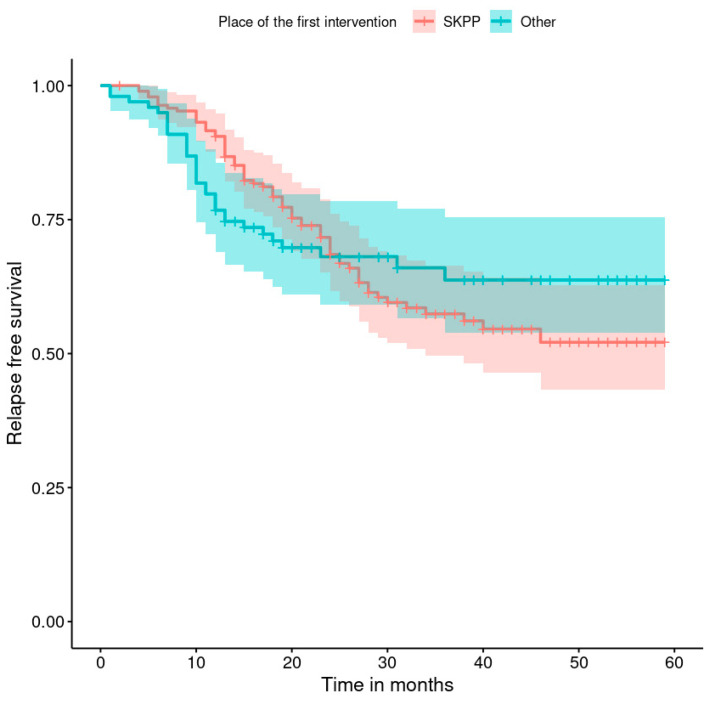
Relapse-free survival curve with 95% CI (darkened area) broken down by the place of the first intervention.

**Figure 3 healthcare-11-02043-f003:**
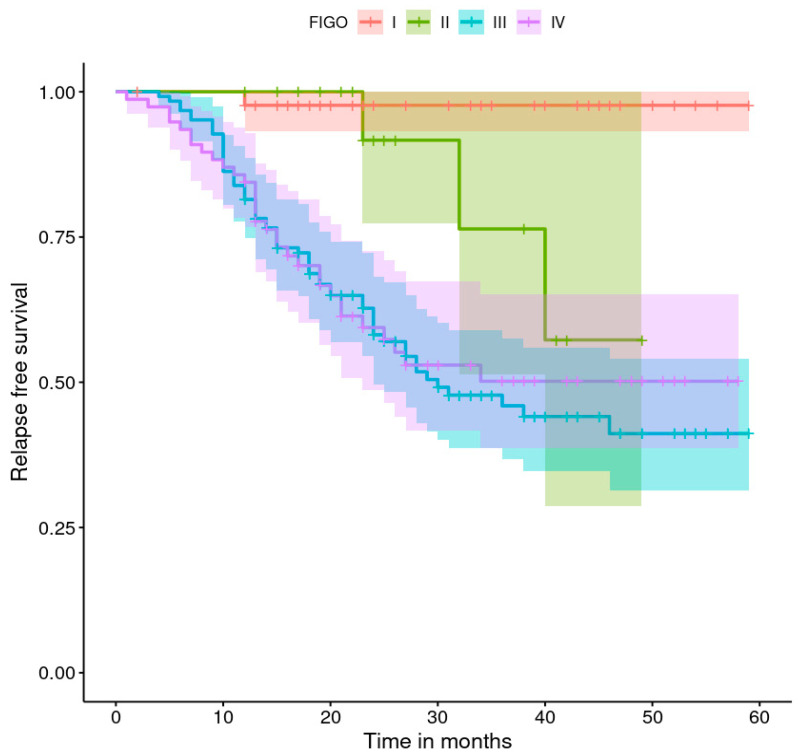
Relapse-free survival curve with 95% CI (darkened area) broken down by the FIGO score.

**Figure 4 healthcare-11-02043-f004:**
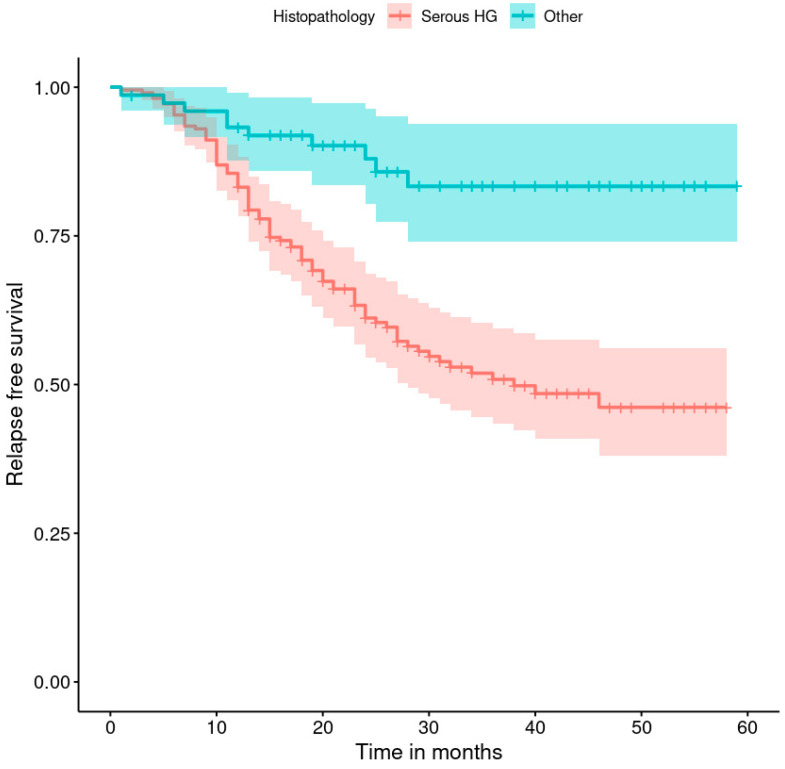
Relapse-free survival curve with 95% CI (darkened area) broken down by the histopathological diagnosis.

**Figure 5 healthcare-11-02043-f005:**
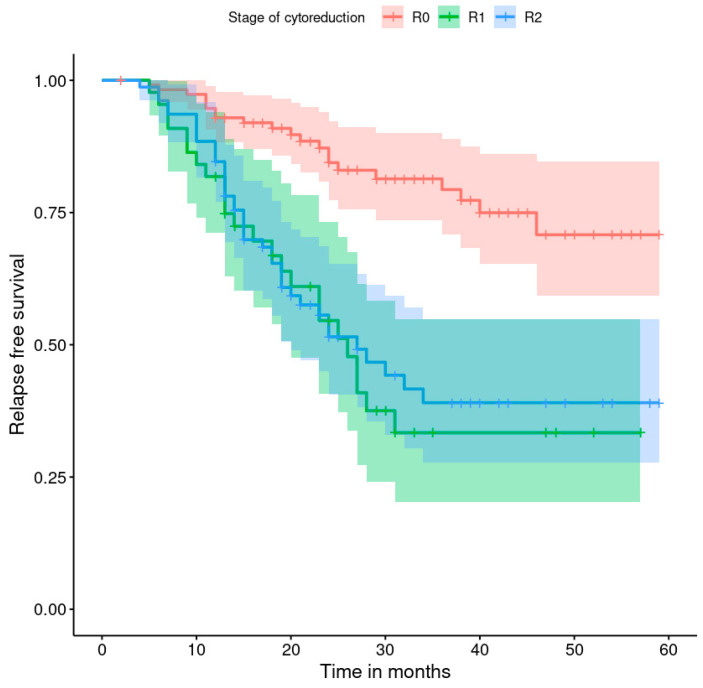
Relapse-free survival curve with 95% CI (darkened area) broken down by the stage of cytoreduction.

**Figure 6 healthcare-11-02043-f006:**
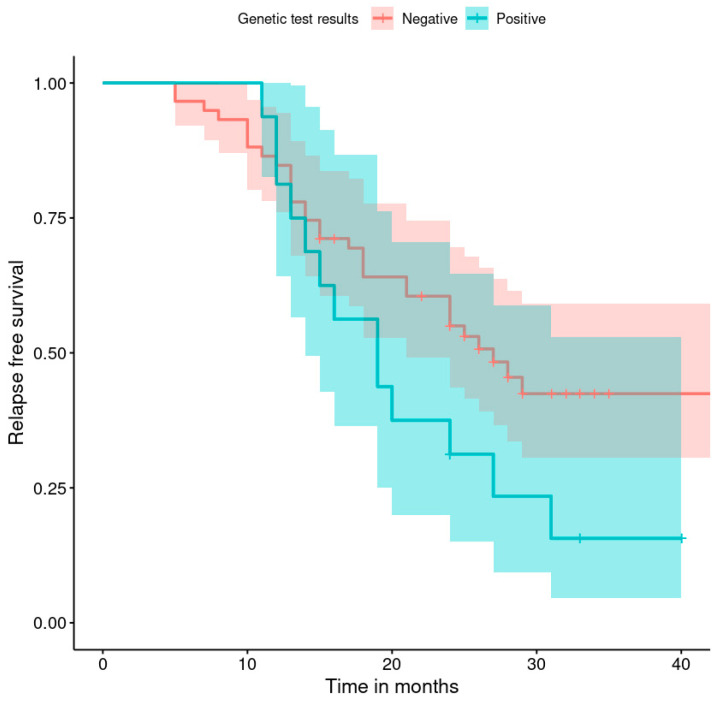
Relapse-free survival curve with 95% CI (darkened area) broken down by the genetic test result.

**Figure 7 healthcare-11-02043-f007:**
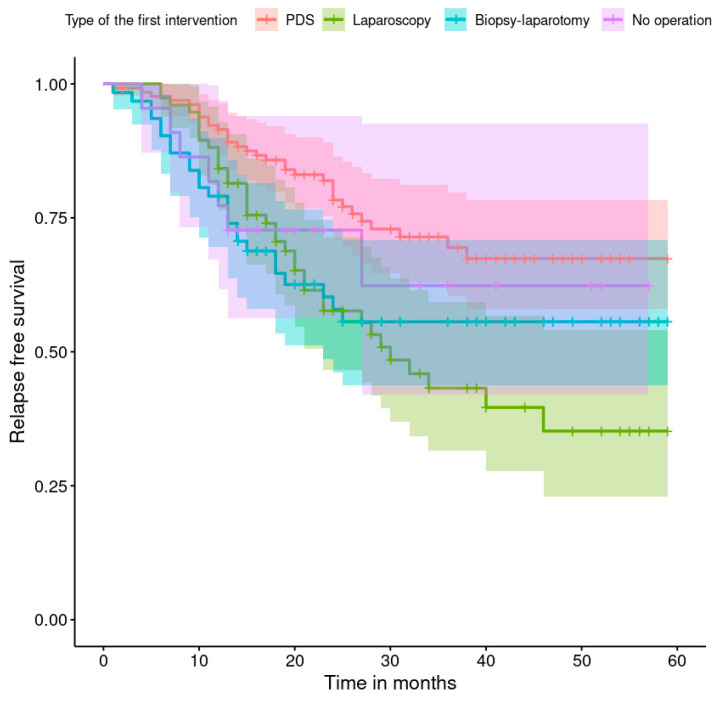
Relapse-free survival curve with 95% CI (darkened area) broken down by the type of the first intervention.

**Figure 8 healthcare-11-02043-f008:**
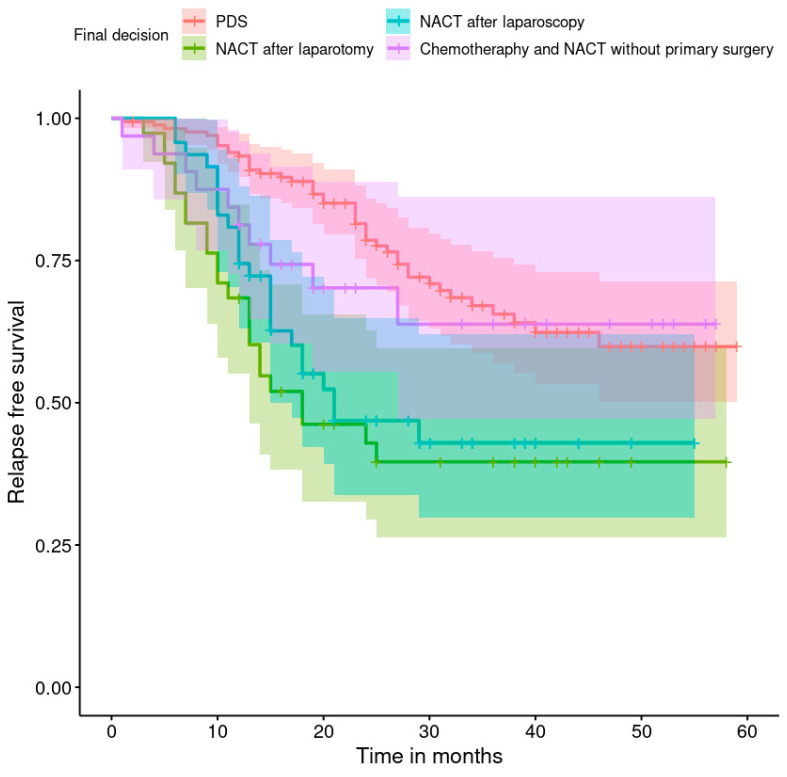
Relapse-free survival curve with 95% CI (darkened area) broken down by the final decision.

**Table 1 healthcare-11-02043-t001:** Baseline characteristics of the study group.

Characteristic	Value, *n* (%)
Year of measurement	
2018	58 (19.9)
2019	66 (22.7)
2020	71 (24.4)
2021	96 (33.0)
Place of first intervention	
SKPP	192 (66.0)
Other hospitals	99 (34.0)
Type of first intervention	
SKPP	
PDS–laparotomy	83 (43.2)
Laparoscopy	70 (36.5)
Biopsy–laparotomy	22 (11.5)
No operation	17 (8.9)
Other hospitals	
PDS–laparotomy	47 (47.5)
Laparoscopy	7 (7.1)
Biopsy–laparotomy	40 (40.4)
No operation	5 (5.1)
Final decision	
PDS after laparotomy or laparoscopy	168 (57.7)
NACT after laparotomy	38 (13.1)
NACT after laparoscopy	47 (16.2)
Chemotherapy	19 (6.5)
NACT without primary surgery	13 (4.5)
Drop-out	2 (0.7)
No intervention	4 (1.4)
FIGO staging system (*n* = 289)	
I	44 (15.2)
II	21 (7.3)
III	125 (43.3)
IV	77 (26.6)
Non-staging	22 (7.6)
Histopathology	
HGSC	215 (74.1)
Other types	75 (25.9)
Stage of cytoreduction (*n* = 237)	
Total (R0)	115 (48.5)
Optimal (R1)	44 (18.6)
Suboptimal (R2)	78 (32.9)
First relapse	100 (34.4)
Second relapse	43 (14.8)
Third relapse	24 (8.2)
Fourth relapse	8 (2.7)
Fifth relapse	3 (1.0)
Sixth relapse	1 (0.3)
Genetic test results (*n* = 75)	
BRCA (−)	59 (78.7)
BRCA (+)	16 (21.3)
Time from first to second relapse (months)—*n* = 43, Me (min–max)	7.00 (1.00; 28.00)
Time from second to third relapse (months)—*n* = 24, Me (min–max)	3.00 (1.00; 11.00)
Time from third to fourth relapse (months)—*n* = 8, Me (min–max)	4.00 (1.00; 12.00)

Me (min–max)—median with range of scores; PDS—primary debulking surgery; LAP—laparoscopy; NACT—neoadjuvant chemotherapy; HGSC—high-grade serous carcinoma.

**Table 2 healthcare-11-02043-t002:** Dependency between the place of first intervention and selected characteristics.

Characteristic, *n* (%)	SKPP Patients	Transferred Patients	Cramer’s V (95% CI)	*p*
FIGO				
I or II	45 (23.7)	20 (20.2)		0.037
III or IV	136 (71.6)	66 (66.7)	0.15 (0.05; 0.28)	
Non-staging	9 (4.7)	13 (13.1)		
Type of first intervention				
PDS–laparotomy	83 (43.2)	47 (47.5)		<0.001
Laparoscopy	70 (36.5)	7 (7.1)	0.41 (0.32; 0.51)	
Exploratory laparotomy	22 (11.5)	40 (40.4)		
No operation	17 (8.9)	5 (5.1)		
Final decision				
PDS after laparoscopy or laparotomy	108 (56.8)	60 (63.2)		<0.001 ^1^
NACT after laparotomy	17 (8.9)	21 (22.1)	0.28 (0.20; 0.38)	
NACT after laparoscopy	43 (22.6)	4 (4.2)		
Chemotherapy	14 (7.4)	5 (5.3)		
NACT without primary surgery	8 (4.2)	5 (5.3)		
Stage of cytoreduction				
Total (R0)	81 (45.5)	34 (57.6)		0.060
Optimal (R1)	31 (17.4)	13 (22.0)	-	
Suboptimal (R2)	66 (37.1)	12 (20.3)		

FIGO—the International Federation of Gynaecology and Obstetrics; PDS—primary debulking surgery; NACT—neoadjuvant chemotherapy; CI—confidence interval; *p*—*p*-value. Dependencies between the groups and selected characteristics were analysed using a chi-squared or Fisher’s exact test ^1^.

**Table 3 healthcare-11-02043-t003:** Comparison of selected characteristics between subjects admitted in the years 2018–2021.

Characteristic, *n* (%)	2018 Group	2019 Group	2020 Group	2021 Group	Cramer’s V (95% CI)	*p*
FIGO staging system						
I or II	11 (19.0)	14 (21.2)	11 (15.9)	29 (30.2)		0.069 ^1^
III or IV	38 (65.5)	49 (74.2)	55 (79.7)	60 (62.5)	-	
Non-staging	9 (15.5)	3 (4.5)	3 (4.3)	7 (7.3)		
Type of first intervention						
PDS–laparotomy	16 (27.6)	30 (45.5)	39 (54.9)	45 (46.9)		0.003 ^1^
Laparoscopy	25 ( )	11 (16.7)	17 (23.9)	24 (25.0)	0.17 (0.14; 0.26)	
Exploratory laparotomy	14 (24.1)	22 (33.3)	11 (15.5)	15 (15.6)		
No operation	3 (5.2)	3 (4.5)	4 (5.6)	12 (12.5)		
Final decision						
PDS after laparoscopy or laparotomy	34 (64.2)	29 (43.9)	41 (57.7)	64 (67.4)		<0.001 ^1^
NACT after laparotomy	4 (7.5)	19 (28.8)	9 (12.7)	6 (6.3)		
NACT after laparoscopy	9 (17.0)	11 (16.7)	17 (23.9)	10 (10.5)	0.27 (0.23; 0.35)	
Chemotherapy	3 (5.7)	6 (9.1)	0 (0.0)	10 (10.5)		
NACT without primary surgery	3 (5.7)	1 (1.5)	4 (5.6)	5 (5.3)		
Place of first intervention						
SKPP	36 (62.1)	42 (63.6)	54 (76.1)	60 (62.5)	-	0.232
Other hospitals	22 (37.9)	24 (36.4)	17 (23.9)	36 (37.5)		
Stage of cytoreduction						
Total (R0)	20 (43.5)	28 (53.8)	26 (41.3)	41 (53.9)		0.308
Optimal (R1)	7 (15.2)	7 (13.5)	18 (28.6)	12 (15.8)	-	
Suboptimal (R2)	19 (41.3)	17 (32.7)	19 (30.2)	23 (30.3)		

FIGO—the International Federation of Gynaecology and Obstetrics; PDS—primary debulking surgery; NACT—neoadjuvant chemotherapy; CI—confidence interval; *p—p*-value. Dependencies between the groups and selected characteristics were analysed using a chi-squared test with Yates’ correction for continuity for two categorical variables or with Fisher’s exact test ^1^.

**Table 4 healthcare-11-02043-t004:** Dependency between the result of the operation, the disease stage, and the place of the first intervention.

Characteristic, *n* (%)	Result of the Operation			*p*
	R0	R1	R2	
FIGO				
I or II	55 (48.2)	1 (2.3)	1 (1.3)	<0.001
III, IV, or no staging	59 (51.8)	43 (97.7)	77 (98.7)	
Place of first intervention				
SKPP	81 (70.4)	31 (70.5)	66 (84.6)	0.060
Other	34 (29.6)	13 (29.5)	12 (15.4)	

Dependencies between the groups and selected characteristics were made using a chi-squared test with Yates’ correction for continuity.

**Table 5 healthcare-11-02043-t005:** Differences in time between relapses among patients from selected groups.

Characteristic	Time from First to Second Relapse (Months), *n* = 43	*p* ^1^	Time from Second to Third Relapse (Months), *n* = 24	*p* ^2^	Time from Third to Fourth Relapse (Months), *n* = 8	*p* ^3^
FIGO						
III	7.00 (2.00; 28.00)	0.788	3.00 (1.00; 5.00)	0.009	4.00 (2.00; 8.00)	-
IV	7.00 (1.00; 16.00)		5.00 (2.00; 11.00)		-	
Place of first intervention						
SKPP	7.00 (2.00; 28.00)	0.306	4.00 (1.00; 11.00)	0.431	4.00 (2.00; 5.00)	0.885
Other	7.00 (1.00; 14.00)		3.00 (1.00; 7.00)		5.50 (1.00; 12.00)	

*p*^1^—*p*-value for the time from first to second relapse; *p*^2^—*p*-value for the time from second to third relapse; *p*^3^—*p*-value for the time from third to fourth relapse.

**Table 6 healthcare-11-02043-t006:** The influence of individual factors on the occurrence of relapse using a univariate Cox regression model.

Variable	HR	95% CI	*p*
Place of first intervention (SKPP vs. other)	0.99	0.65–1.51	0.958
FIGO (I or II vs. III, IV, or no staging)	8.49	3.12–23.10	<0.001
Histopathology (HGSC vs. other)	0.26	0.14–0.51	<0.001
Type of first intervention (vs. PDS–laparotomy)			
Laparoscopy	2.28	1.38–3.59	0.001
Exploratory laparotomy	1.92	1.14–3.24	0.015
No operation	1.59	0.70–3.60	0.267
Type of first intervention (vs. laparoscopy)			
Exploratory laparotomy	0.87	0.52–1.45	0.595
No operation	0.71	0.32–1.60	0.411
Final decision (vs. PDS)			
NACT after laparotomy	2.97	1.78–4.52	<0.001
NACT after laparoscopy	2.52	1.53–4.14	<0.001
Chemotherapy and NACT without primary surgery	1.40	0.70–2.77	0.342
Stage of cytoreduction (vs. total (R0))			
Optimal (R1)	4.02	2.23–7.25	<0.001
Suboptimal (R2)	3.55	2.08–6.05	<0.001
Genetic test results (+ vs. −)	1.79	0.93–3.42	0.081

FIGO—the International Federation of Gynaecology and Obstetrics; HGSC—high-grade serous carcinoma; PDS—primary debulking surgery; NACT—neoadjuvant chemotherapy; *p*—*p*-value; HR and 95% CI—hazard ratio with a 95% confidence interval for the occurrence of relapse.

**Table 7 healthcare-11-02043-t007:** The influence of individual factors on the occurrence of relapse using the univariate Cox regression model.

Variable	HR	95% CI	*p*
FIGO (I or II vs. III, IV, or no staging)	2.75	0.92–8.23	0.071
Histopathology (HGSC vs. other)	0.46	0.19–1.10	0.079
Final decision (vs. PDS)			
NACT after laparotomy	2.81	1.58–4.98	<0.001
NACT after laparoscopy	1.99	1.17–3.39	0.011
Chemotherapy and NACT without primary surgery	1.21	0.47–3.11	0.696
Stage of cytoreduction (vs. total (R0))			
Optimal (R1)	2.43	1.30–4.56	0.005
Suboptimal (R2)	1.96	1.11–3.46	0.020

FIGO—the International Federation of Gynaecology and Obstetrics; HGSC—high-grade serous carcinoma; PDS—primary debulking surgery; NACT—neoadjuvant chemotherapy; *p*—*p*-value; HR and 95% CI—hazard ratio with a 95% confidence interval for the occurrence of relapse.

## Data Availability

All data is available at a corresponding author.

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
