# Peer review of "Epithelial Ovarian Cancer—Varied Treatment Results"

_healthcare, 2023, doi:10.3390/healthcare11142043_

Round 1

Reviewer 1 Report

In this manuscript, the authors revealed the correlation between treatment and survival in ovarian cancer in Poland through a retrospective observational study enrolled 289 patients, and proposed the necessity of identifying new markers for detecting FIGO I and II OC and conducting comprehensive histopathological and genetic tests. This is a study with a certain significance, but some points are required to be improved.

1. In the abstract, some acronyms such as SKPP, NACT, PFS, PDS were not defined when they appeared for the first time.

2. In this manuscript, the authors only described the inclusion and exclusion criteria in the Materials and Methods section, which is too simple. Information such as the study group, the different treatments the patients received, data collection and collation and the statistical methods used for data analysis also need to be explained in detail.

3. There seems to be something wrong with the format shown in Table 1.

4. The single-center clinical study has certain biases and limitations. I hope that the authors can further expand the sample size and conduct multi-center studies in the future.

5. The short observation period is an important limitation of this study. It is hoped that the authors can continue to follow up these patients to obtain more comprehensive data and more reliable results.

6. Most of an article's references should be to primary research from the past 2–5 years. The references of this article are too old. It is recommended to update them in time.

None.

Author Response

We thank the Reviewer for taking the time to read the paper and providing comments that improve the quality of our work.

Please see our comments in the attached file.

Reviewer 2 Report

1. The purpose of the abstract is to give brief information about the content of the article. This abstract is difficult to read because of the many abbreviations that are not explained. Without reading the entire article, it is very difficult to understand the meaning of the abstract.

2. The term ovarian cancer is broad. It is unclear whether the authors studied malignant epithelial tumors of the ovaries (ovarian carcinoma ) or all other malignant tumor types. In my opinion, it is incorrect to analyze all malignant ovarian tumors together because they are completely different diseases with different pathogenesis, molecular alterations, clinical course, and prognosis. As the authors noted, most of the tumors studied were high-grade serous carcinomas. In my opinion, such a study should be performed for high-grade serous carcinomas.

3. Too many curves (figures) in the results are confusing.

4. The conclusions are general, the need for early detection of ovarian cancer does not follow directly from this study.

Author Response

(The authors gave the same response as above.)

Reviewer 3 Report

The paper titled "Epithelial ovarian cancer – varied treatment results" presents an investigation into the treatment outcomes of epithelial ovarian cancer. The study appears to be valuable in providing insights into the varied responses to treatment. However, there are several points that need to be addressed before the paper can be considered for publication:

1.The paper lacks a clear description of the research design and methodology employed. It is crucial to provide information on the study design (e.g., retrospective, prospective), the sample size, age, and interventions investigated. Additionally, it would be helpful to discuss any potential biases or limitations associated with the retrospective design of the study.

2.The paper presents some findings regarding treatment outcomes in epithelial ovarian cancer. However, the results section is brief and lacks comprehensive reporting. It is important to provide detailed information on the treatment response rates, and overall survival, including appropriate statistical measures (e.g., hazard ratios, confidence intervals). Presenting the results in a clear and organized manner will enhance the readers' understanding of the study findings.

3.While the statistical analysis conducted is generally appropriate, more information is required to fully understand the results. The paper should provide details on the statistical tests used in the method part, including the rationale for choosing each test and the assumptions made. Furthermore, it would be beneficial to report effect sizes or confidence intervals for the significant associations identified.

other minor comments:

1. Page 3, line 87, "admitted" should be "were admitted"

2. Page 4, Table 1 title was missing

3. Page 6, line 156, "presents" should be "presented in"

4. Page 7, line 167, "shows" should be"shown in".

5. Page 8, line 182, "second" should be"the second".

6. Page 10, lines 241, 242, and 251, "presents" should be "presented".

The manuscript would benefit from additional proofreading to improve grammar, sentence structure, and overall readability. Additionally, consider reorganizing the sections to improve the flow and logical progression of ideas.

Author Response

(The authors gave the same response as above.)

Round 2

Reviewer 2 Report

The revised manuscript is acceptable for publishing.

Author Response

Thank you very much

Reviewer 3 Report

The author answers well of the questions and agrees to accept.

Author Response

Thank You very much.